# PREFERENCES IMPLICIT IN THE STATE OF THE WORLD

**Rohin Shah** [* †]
UC Berkeley

**Dmitrii Krasheninnikov** [* † ‡]
University of Amsterdam

**Jordan Alexander** [† ‡]
Stanford University

**Pieter Abbeel**
UC Berkeley

**Anca D. Dragan**
UC Berkeley

## ABSTRACT

Reinforcement learning (RL) agents optimize only the features specified in a reward function and are indifferent to anything left out inadvertently. This means that we must not only specify what *to* do, but also the much larger space of what *not* to do. It is easy to forget these preferences, since these preferences are *already* satisfied in our environment. This motivates our key insight: *when a robot is deployed in an environment that humans act in, the state of the environment is already optimized for what humans want*. We can therefore use this implicit preference information from the state to fill in the blanks. We develop an algorithm based on Maximum Causal Entropy IRL and use it to evaluate the idea in a suite of proof-of-concept environments designed to show its properties. We find that information from the initial state can be used to infer both side effects that should be avoided as well as preferences for how the environment should be organized. Our code can be found at `https://github.com/HumanCompatibleAI/rlsp`.

## 1 INTRODUCTION

Deep reinforcement learning (deep RL) has been shown to succeed at a wide variety of complex tasks given a correctly specified reward function. Unfortunately, for many real-world tasks it can be challenging to specify a reward function that captures human preferences, particularly the preference for avoiding unnecessary side effects while still accomplishing the goal (Amodei et al., 2016). As a result, there has been much recent work (Christiano et al., 2017; Fu et al., 2017; Sadigh et al., 2017) that aims to learn specifications for tasks a robot should perform.

Typically when learning about what people want and don't want, we look to human action as evidence: what reward they specify (Hadfield-Menell et al., 2017), how they perform a task (Ziebart et al., 2010; Fu et al., 2017), what choices they make (Christiano et al., 2017; Sadigh et al., 2017), or how they rate certain options (Daniel et al., 2014). Here, we argue that there is an additional source of information that is potentially rather helpful, but that we have been ignoring thus far:

> *The key insight of this paper is that when a robot is deployed in an environment that humans have been acting in, the state of the environment is already optimized for what humans want.*

For example, consider an environment in which a household robot must navigate to a goal location without breaking any vases in its path, illustrated in Figure 1. The human operator, Alice, asks the robot to go to the purple door, forgetting to specify that it should also avoid breaking vases along the way. However, since the robot has been deployed in a state that only contains unbroken vases, it can infer that while acting in the environment (prior to robot's deployment), Alice was using one of the relatively few policies that do not break vases, and so must have cared about keeping vases intact.

---

[*]equal contribution

[†]`rohinmshah@berkeley.edu`, `dmitrii.krasheninnikov@student.uva.nl`, `jfalex@stanford.edu`

[‡]work done at UC Berkeley

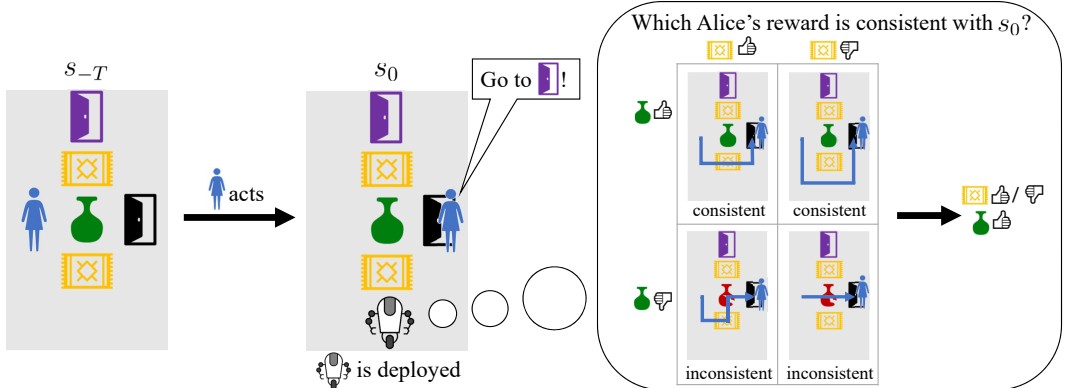

Figure 1: An illustration of learning preferences from an initial state. Alice attempts to accomplish a goal in an environment with an easily breakable vase in the center. The robot observes the state of the environment, $s_0$, after Alice has acted for some time from an even earlier state $s_{-T}$. It considers multiple possible human reward functions, and infers that states where vases are intact usually occur when Alice's reward penalizes breaking vases. In contrast, it doesn't matter much what the reward function says about carpets, as we would observe the same final state either way. Note that while we consider a specific $s_{-T}$ for clarity here, the robot could also reason using a distribution over $s_{-T}$.

The initial state $s_0$ can contain information about arbitrary preferences, including tasks that the robot should actively perform. For example, if the robot observes a basket full of apples near an apple tree, it can reasonably infer that Alice wants to harvest apples. However, $s_0$ is particularly useful for inferring which side effects humans care about. Recent approaches avoid unnecessary side effects by penalizing changes from an inaction baseline (Krakovna et al., 2018; Turner, 2018). However, this penalizes *all* side effects. The inaction baseline is appealing precisely because the initial state has already been optimized for human preferences, and action is more likely to ruin $s_0$ than inaction. If our robot infers preferences from $s_0$, it can avoid negative side effects while allowing positive ones.

This work is about highlighting the potential of this observation, and as such makes unrealistic assumptions, such as known dynamics and hand-coded features. Given just $s_0$, these assumptions are necessary: without dynamics, it is hard to tell whether some feature of $s_0$ was created by humans or not. Nonetheless, we are optimistic that these assumptions can be relaxed, so that this insight can be used to improve deep RL systems. We suggest some approaches in our discussion.

Our contributions are threefold. First, we identify the state of the world at initialization as a source of information about human preferences. Second, we leverage this insight to derive an algorithm, Reward Learning by Simulating the Past (RLSP), which infers reward from initial state based on a Maximum Causal Entropy (Ziebart et al., 2010) model of human behavior. Third, we demonstrate the properties and limitations of RLSP on a suite of proof-of-concept environments: we use it to avoid side effects, as well as to learn implicit preferences that require active action. In Figure 1 the robot moves to the purple door without breaking the vase, despite the lack of a penalty for breaking vases.

## 2 RELATED WORK

**Preference learning.** Much recent work has learned preferences from different sources of data, such as demonstrations (Ziebart et al., 2010; Ramachandran and Amir, 2007; Ho and Ermon, 2016; Fu et al., 2017; Finn et al., 2016), comparisons (Christiano et al., 2017; Sadigh et al., 2017; Wirth et al., 2017), ratings (Daniel et al., 2014), human reinforcement signals (Knox and Stone, 2009; Warnell et al., 2017; MacGlashan et al., 2017), proxy rewards (Hadfield-Menell et al., 2017), etc. We suggest preference learning with a new source of data: the state of the environment when the robot is first deployed. It can also be seen as a variant of Maximum Causal Entropy Inverse Reinforcement Learning (Ziebart et al., 2010): while inverse reinforcement learning (IRL) requires demonstrations, or at least state sequences without actions (Edwards et al., 2018; Yu et al., 2018), we learn a reward function from a single state, albeit with the simplifying assumption of known dynamics. This can also be seen as an instance of IRL from summary data (Kangasrääsiö and Kaski, 2018).

**Frame properties.** The frame problem in AI (McCarthy and Hayes, 1981) refers to the issue that we must specify what stays the same in addition to what changes. In formal verification, this manifests as a requirement to explicitly specify the many quantities that the program does not change (Andreescu, 2017). Analogously, rewards are likely to specify what to do (the task), but may forget to say what *not* to do (the frame properties). One of our goals is to infer frame properties automatically.

**Side effects.** An impact penalty can mitigate reward specification problems, since it penalizes unnecessary "large" changes (Armstrong and Levinstein, 2017). We could penalize a reduction in the number of reachable states (Krakovna et al., 2018) or attainable utility (Turner, 2018). However, such approaches will penalize all irreversible effects, including ones that humans *want*. In contrast, by taking a preference inference approach, we can infer which effects humans care about.

**Goal states as specifications.** Desired behavior in RL can be specified with an explicitly chosen goal state (Kaelbling, 1993; Schaul et al., 2015; Nair et al., 2018; Bahdanau et al., 2018; Andrychowicz et al., 2017). In our setting, the robot observes the *initial* state $s_0$ where it *starts* acting, which is not explicitly chosen by the designer, but nonetheless contains preference information.

## 3 PRELIMINARIES

A finite-horizon Markov decision process (MDP) is a tuple $\mathcal{M} = \langle \mathcal{S}, \mathcal{A}, \mathcal{T}, r, T \rangle$, where $\mathcal{S}$ is the set of states, $\mathcal{A}$ is the set of actions, $\mathcal{T} : \mathcal{S} \times \mathcal{A} \times \mathcal{S} \mapsto [0, 1]$ is the transition probability function, $r : \mathcal{S} \mapsto \mathbb{R}$ is the reward function, and $T \in \mathbb{Z}_+$ is the finite planning horizon. We consider MDPs where the reward is linear in features, and does not depend on action: $r(s; \theta) = \theta^T f(s)$, where $\theta$ are the parameters defining the reward function and $f$ computes features of a given state.

**Inverse Reinforcement Learning (IRL).** In IRL, the aim is to infer the reward function $r$ given an MDP without reward $\mathcal{M} \backslash r$ and expert demonstrations $\mathcal{D} = \{\tau_1, ..., \tau_n\}$, where each $\tau_i = (s_0, a_0, ..., s_T, a_T)$ is a trajectory sampled from the expert policy acting in the MDP. It is assumed that each $\tau_i$ is feasible, so that $\mathcal{T}(s_{j+1} \mid s_j, a_j) > 0$ for every $j$.

**Maximum Causal Entropy IRL (MCEIRL).** As human demonstrations are rarely optimal, Ziebart et al. (2010) models the expert as a Boltzmann-rational agent that maximizes total reward and causal entropy of the policy. This leads to the policy $\pi_t(a \mid s, \theta) = \exp(Q_t(s, a; \theta) - V_t(s; \theta))$, where $V_t(s; \theta) = \ln \sum_a exp(Q_t(s, a; \theta))$ plays the role of a normalizing constant. Intuitively, the expert is assumed to act close to randomly when the difference in expected total reward across the actions is small, but nearly always chooses the best action when it leads to a substantially higher expected return. The soft Bellman backup for the state-action value function $Q$ is the same as usual, and is given by $Q_t(s, a; \theta) = \theta^T f(s) + \sum_{s'} \mathcal{T}(s' \mid s, a) V_{t+1}(s'; \theta)$.

The likelihood of a trajectory $\tau$ given the reward parameters $\theta$ is:

$$p(\tau \mid \theta) = p(s_0) \bigg( \prod_{t=0}^{T-1} \mathcal{T}(s_{t+1} \mid s_t, a_t) \pi_t(a_t \mid s_t, \theta) \bigg) \pi_T(a_T \mid s_T, \theta). \tag{1}$$

MCEIRL finds the reward parameters $\theta^*$ that maximize the log-likelihood of the demonstrations:

$$\theta^* = \text{argmax}_\theta \ln p(\mathcal{D} \mid \theta) = \text{argmax}_\theta \sum_i \sum_t \ln \pi_t(a_{i,t} \mid s_{i,t}, \theta). \tag{2}$$

$\theta^*$ gives rise to a policy whose feature expectations match those of the expert demonstrations.

## 4 REWARD LEARNING BY SIMULATING THE PAST

We solve the problem of learning the reward function of an expert Alice given a single final state of her trajectory; we refer to this problem as *IRL from a single state*. Formally, we aim to infer Alice's reward $\theta$ given an environment $\mathcal{M} \backslash r$ and the last state of the expert's trajectory $s_0$.

**Formulation.** To adapt MCEIRL to the one state setting we modify the observation model from Equation 1. Since we only have a single end state $s_0$ of the trajectory $\tau_0 = (s_{-T}, a_{-T}, ..., s_0, a_0)$, we marginalize over all of the other variables in the trajectory:

$$p(s_0 \mid \theta) = \sum_{s_{-T}, a_{-T}, ... s_{-1}, a_{-1}, a_0} p(\tau_0 \mid \theta), \tag{3}$$

where $p(\tau_0 \mid \theta)$ is given in Equation 1. We could invert this and sample from $p(\theta \mid s_0)$; the resulting algorithm is presented in Appendix C, but is relatively noisy and slow. We instead find the MLE:

$$\theta^* = \mathrm{argmax}_\theta \ln p(s_0 \mid \theta). \tag{4}$$

**Solution.** Similarly to MCEIRL, we use a gradient ascent algorithm to solve the IRL from one state problem. We explain the key steps here and give the full derivation in Appendix B. First, we express the gradient in terms of the gradients of trajectories:

$$\nabla_\theta \ln p(s_0 \mid \theta) = \sum_{\tau_{-T:-1}} p(\tau_{-T:-1} \mid s_0, \theta) \nabla_\theta \ln p(\tau_{-T:0} \mid \theta).$$

This has a nice interpretation – compute the Maximum Causal Entropy gradients for each trajectory, and then take their weighted sum, where each weight is the probability of the trajectory given the evidence $s_0$ and current reward $\theta$. We derive the exact gradient for a trajectory instead of the approximate one in Ziebart et al. (2010) in Appendix A and substitute it in to get:

$$\nabla_\theta \ln p(s_0) = \frac{1}{p(s_0)} \sum_{\tau_{-T:-1}} \left[ p(\tau_{-T:-1}, s_0) \sum_{t=-T}^{-1} \left( f(s_t) + \mathbb{E}_{s'_{t+1}} \left[ \mathcal{F}_{t+1}(s'_{t+1}) \right] - \mathcal{F}_t(s_t) \right) \right], \tag{5}$$

where we have suppressed the dependence on $\theta$ for readability. $\mathcal{F}_t(s_t)$ denotes the expected features when starting at $s_t$ at time $t$ and acting until time 0 under the policy implied by $\theta$.

Since we combine gradients from simulated past trajectories, we name our algorithm Reward Learning by Simulating the Past (RLSP). The algorithm computes the gradient using dynamic programming, detailed in Appendix B. We can easily incorporate a prior on $\theta$ by adding the gradient of the log prior to the gradient in Equation 5.

## 5 EVALUATION

Evaluation of RLSP is non-trivial. The inferred reward is very likely to assign state $s_0$ maximal reward, since it was inferred under the assumption that when Alice optimized the reward she ended up at $s_0$. If the robot then starts in state $s_0$, if a no-op action is available (as it often is), the RLSP reward is likely to incentivize no-ops, which is not very interesting.

Ultimately, we hope to use RLSP to correct badly specified instructions or reward functions. So, we created a suite of environments with a true reward $R_{\mathrm{true}}$, a specified reward $R_{\mathrm{spec}}$, Alice's first state $s_{-T}$, and the robot's initial state $s_0$, where $R_{\mathrm{spec}}$ ignores some aspect(s) of $R_{\mathrm{true}}$. RLSP is used to infer a reward $\theta_{\mathrm{Alice}}$ from $s_0$, which is then combined with the specified reward to get a final reward $\theta_{\mathrm{final}} = \theta_{\mathrm{Alice}} + \lambda \theta_{\mathrm{spec}}$. (We considered another method for combining rewards; see Appendix D for details.) We inspect the inferred reward qualitatively and measure the expected amount of true reward obtained when planning with $\theta_{\mathrm{final}}$, as a fraction of the expected true reward from the optimal policy. We tune the hyperparameter $\lambda$ controlling the tradeoff between $R_{\mathrm{spec}}$ and the human reward for all algorithms, including baselines. We use a Gaussian prior over the reward parameters.

### 5.1 BASELINES

**Specified reward policy $\pi_{\mathbf{spec}}$.** We act as if the true reward is exactly the specified reward.

**Policy that penalizes deviations $\pi_{\mathbf{deviation}}$.** This baseline minimizes change by penalizing deviations from the observed features $f(s_0)$, giving $R_{\mathrm{final}}(s) = \theta_{\mathrm{spec}}^T f(s) + \lambda ||f(s) - f(s_0)||$.

**Relative reachability policy $\pi_{\mathbf{reachability}}$.** Relative reachability (Krakovna et al., 2018) considers a change to be negative when it decreases *coverage*, relative to what would have happened had the agent done nothing. Here, coverage is a measure of how easily states can be reached from the current state. We compare against the variant of relative reachability that uses undiscounted coverage and a baseline policy where the agent takes no-op actions, as in the original paper. Relative reachability requires known dynamics but not a handcoded featurization. A version of relative reachability that operates in feature space instead of state space would behave similarly.

## 5.2 COMPARISON TO BASELINES

We compare RLSP to our baselines with the assumption of known $s_{-T}$, because it makes it easier to analyze RLSP's properties. We consider the case of unknown $s_{-T}$ in Section 5.3. We summarize the results in Table 1, and show the environments and trajectories in Figure 2.

Table 1: Performance of algorithms on environments designed to test particular properties.

| | Side effects Room | Env effect Toy train | Implicit reward Apple collection | Desirable effect Batteries | | Unseen effect Far away vase |
| --- | :---: | :---: | :---: | :---: | :---: | :---: |
| | | | | Easy | Hard | |
| $\pi_{\text{spec}}$ | ✗ | ✗ | ✗ | ✓ | ✗ | ✗ |
| $\pi_{\text{deviation}}$ | ✓ | ✗ | ✗ | ≈ | ✗ | ✓ |
| $\pi_{\text{reachability}}$ | ✓ | ✓ | ✗ | ≈ | ✗ | ✓ |
| $\pi_{\text{RLSP}}$ | ✓ | ✓ | ✓ | ✓ | ✓ | ✗ |

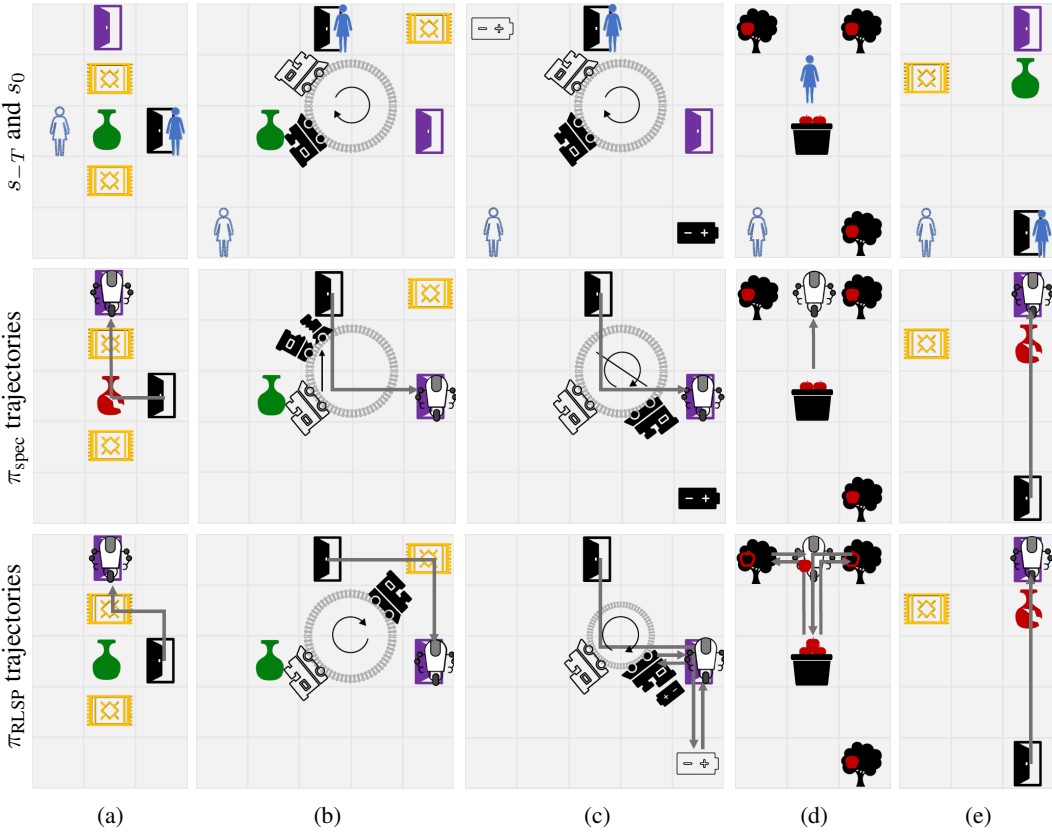

(a)    (b)    (c)    (d)    (e)

Figure 2: Evaluation of RLSP on our environments. Silhouettes indicate the initial position of an object or agent, while filled in version indicate their positions after an agent has acted. The first row depicts the information given to RLSP. The second row shows the trajectory taken by the robot when following the policy $\pi_{\text{spec}}$ that is optimal for $\theta_{\text{spec}}$. The third row shows the trajectory taken when following the policy $\pi_{\text{RLSP}}$ that is optimal for $\theta_{\text{final}} = \theta_{\text{Alice}} + \lambda\theta_{\text{spec}}$. (a) Side effects: Room with vase (b) Distinguishing environment effects: Toy train (c) Implicit reward: Apple collection (d) Desirable side effect: Batteries (e) "Unseen" side effect: Room with far away vase.

**Side effects: Room with vase** *(Figure 2a)*. The room tests whether the robot can avoid breaking a vase as a side effect of going to the purple door. There are features for the number of broken vases, standing on a carpet, and each door location. Since Alice didn't walk over the vase, RLSP infers a negative reward on broken vases, and a small positive reward on carpets (since paths to the top door usually involve carpets). So, $\pi_{\text{RLSP}}$ successfully avoids breaking the vase. The penalties also achieve

the desired behavior: $\pi_{\text{deviation}}$ avoids breaking the vase since it would change the "number of broken vases" feature, while relative reachability avoids breaking the vase since doing so would result in all states with intact vases becoming unreachable.

**Distinguishing environment effects: Toy train** *(Figure 2b)*. To test whether algorithms can distinguish between effects caused by the agent and effects caused by the environment, as suggested in Krakovna et al. (2018), we add a toy train that moves along a predefined track. The train breaks if the agent steps on it. We add a new feature indicating whether the train is broken and new features for each possible train location. As before, the specified reward only has a positive weight on the purple door, while the true reward also penalizes broken trains and vases.

RLSP infers a negative reward on broken vases and broken trains, for the same reason as before. It also infers not to put any weight on any particular train location, even though it changes frequently, because it doesn't help explain $s_0$. As a result, $\pi_{\text{RLSP}}$ walks over a carpet, but not a vase or a train. $\pi_{\text{deviation}}$ immediately breaks the train to keep the train location the same. $\pi_{\text{reachability}}$ deduces that breaking the train is irreversible, and so follows the same trajectory as $\pi_{\text{RLSP}}$.

**Implicit reward: Apple collection** *(Figure 2d)*. This environment tests whether the algorithms can learn tasks implicit in $s_0$. There are three trees that grow apples, as well as a basket for collecting apples, and the goal is for the robot to harvest apples. However, the specified reward is zero: the robot must infer the task from the observed state. We have features for the number of apples in baskets, the number of apples on trees, whether the robot is carrying an apple, and each location that the agent could be in. $s_0$ has two apples in the basket, while $s_{-T}$ has none.

$\pi_{\text{spec}}$ is arbitrary since every policy is optimal for the zero reward. $\pi_{\text{deviation}}$ does nothing, achieving zero reward, since its reward can never be positive. $\pi_{\text{reachability}}$ also does not harvest apples. RLSP infers a positive reward on apples in baskets, a negative reward for apples on trees, and a small positive reward for carrying apples. Despite the spurious weights, $\pi_{\text{RLSP}}$ harvests apples as desired.

**Desirable side effect: Batteries** *(Figure 2c)*. This environment tests whether the algorithms can tell when a side effect is allowed. We take the toy train environment, remove vases and carpets, and add batteries. The robot can pick up batteries and put them into the (now unbreakable) toy train, but the batteries are never replenished. If the train runs for 10 timesteps without a new battery, it stops operating. There are features for the number of batteries, whether the train is operational, each train location, and each door location. There are two batteries at $s_{-T}$ but only one at $s_0$. The true reward incentivizes an operational train and being at the purple door. We consider two variants for the task reward – an "easy" case, where the task reward equals the true reward, and a "hard" case, where the task reward only rewards being at the purple door.

Unsurprisingly, $\pi_{\text{spec}}$ succeeds at the easy case, and fails on the hard case by allowing the train to run out of power. Both $\pi_{\text{deviation}}$ and $\pi_{\text{reachability}}$ see the action of putting a battery in the train as a side effect to be penalized, and so neither can solve the hard case. They penalize picking up the batteries, and so only solve the easy case if the penalty weight is small. RLSP sees that one battery is gone and that the train is operational, and infers that Alice wants the train to be operational and doesn't want batteries (since a preference against batteries and a preference for an operational train are nearly indistinguishable). So, it solves both the easy and the hard case, with $\pi_{\text{RLSP}}$ picking up the battery, then staying at the purple door except to deliver the battery to the train.

**"Unseen" side effect: Room with far away vase** *(Figure 2e)*. This environment demonstrates a limitation of our algorithm: it cannot identify side effects that Alice would never have triggered. In this room, the vase is nowhere close to the shortest path from the Alice's original position to her goal, but is on the path to the robot's goal. Since our baselines don't care about the trajectory the human takes, they all perform as before: $\pi_{\text{spec}}$ walks over the vase, while $\pi_{\text{deviation}}$ and $\pi_{\text{reachability}}$ both avoid it. Our method infers a near zero weight on the broken vase feature, since it is not present on any reasonable trajectory to the goal, and so breaks it when moving to the goal. Note that this only applies when Alice is known to be at the bottom left corner at $s_{-T}$: if we have a uniform prior over $s_{-T}$ (considered in Section 5.3) then we do consider trajectories where vases are broken.

## 5.3 Comparison between knowing $s_{-T}$ vs. a distribution over $s_{-T}$

So far, we have considered the setting where the robot knows $s_{-T}$, since it is easier to analyze what happens. However, typically we will not know $s_{-T}$, and will instead have some prior over $s_{-T}$. Here,

we compare RLSP in two settings: perfect knowledge of $s_{-T}$ (as in Section 5.2), and a uniform distribution over all states.

**Side effects: Room with vase** *(Figure 2a)* **and toy train** *(Figure 2b).* In both room with vase and toy train, RLSP learns a smaller negative reward on broken vases when using a uniform prior. This is because RLSP considers many more feasible trajectories when using a uniform prior, many of which do not give Alice a chance to break the vase, as in Room with far away vase in Section 5.2. In room with vase, the small positive reward on carpets changes to a near-zero negative reward on carpets. With known $s_{-T}$, RLSP overfits to the few consistent trajectories, which usually go over carpets, whereas with a uniform prior it considers many more trajectories that often don't go over carpets, and so it correctly infers a near-zero weight. In toy train, the negative reward on broken trains becomes slightly more negative, while other features remain approximately the same. This may be because when Alice starts out closer to the toy train, she has more of an opportunity to break it, compared to the known $s_{-T}$ case.

**Implicit preference: Apple collection** *(Figure 2d).* Here, a uniform prior leads to a smaller positive weight on the number of apples in baskets compared to the case with known $s_{-T}$. Intuitively, this is because RLSP is considering cases where $s_{-T}$ already has one or two apples in the basket, which implies that Alice has collected fewer apples and so must have been less interested in them. States where the basket starts with three or more apples are inconsistent with the observed $s_0$ and so are not considered. Following the inferred reward still leads to good apple harvesting behavior.

**Desirable side effects: Batteries** *(Figure 2c).* With the uniform prior, we see the same behavior as in Apple collection, where RLSP with a uniform prior learns a slightly smaller negative reward on the batteries, since it considers states $s_{-T}$ where the battery was already gone. In addition, due to the particular setup the battery must have been given to the train two timesteps prior, which means that in any state where the train started with very little charge, it was allowed to die even though a battery could have been provided before, leading to a near-zero *positive* weight on the train losing charge. Despite this, RLSP successfully delivers the battery to the train in both easy and hard cases.

**"Unseen" side effect: Room with far away vase** *(Figure 2e).* With a uniform prior, we "see" the side effect: if Alice started at the purple door, then the shortest trajectory to the black door would break a vase. As a result, $\pi_{\text{RLSP}}$ successfully avoids the vase (whereas it previously did not). Here, uncertainty over the initial state $s_{-T}$ can counterintuitively *improve* the results, because it increases the diversity of trajectories considered, which prevents RLSP from "overfitting" to the few trajectories consistent with a known $s_{-T}$ and $s_0$.

Overall, RLSP is quite robust to the use of a uniform prior over $s_{-T}$, suggesting that we do not need to be particularly careful in the design of that prior.

## 5.4 ROBUSTNESS TO THE CHOICE OF ALICE'S PLANNING HORIZON

We investigate how RLSP performs when assuming the wrong value of Alice's planning horizon $T$. We vary the value of $T$ assumed by RLSP, and report the true return achieved by $\pi_{\text{RLSP}}$ obtained using the inferred reward and a *fixed* horizon for the robot to act. For this experiment, we used a uniform prior over $s_{-T}$, since with known $s_{-T}$, RLSP often detects that the given $s_{-T}$ and $s_0$ are incompatible (when $T$ is misspecified). The results are presented in Figure 3.

The performance worsens when RLSP assumes that Alice had a smaller planning horizon than she actually had. Intuitively, if we assume that Alice has only taken one or two actions ever, then even if we knew the actions they could have been in service of many goals, and so we end up quite uncertain about Alice's reward.

When the assumed $T$ is larger than the true horizon, RLSP correctly infers things the robot should *not* do. Knowing that the vase was not broken for longer than $T$ timesteps is more evidence to suspect that Alice cared about not breaking the vase. However, overestimated $T$ leads to worse performance at inferring implicit preferences, as in the Apples environment. If we assume Alice has only collected two apples in 100 timesteps, she must not have cared about them much, since she could have collected many more. The batteries

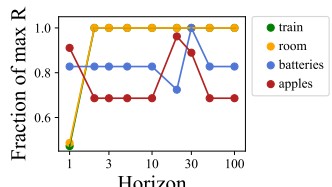

Figure 3: Reward achieved by $\pi_{\text{RLSP}}$, as a fraction of the expected reward of the optimal policy, for different values of Alice's planning horizon $T$.

environment is unusual – assuming that Alice has been acting for 100 timesteps, the only explanation for the observed $s_0$ is that Alice waited until the 98th timestep to put the battery into the train. This is not particularly consistent with any reward function, and performance degrades.

Overall, $T$ is an important parameter and needs to be set appropriately. However, even when $T$ is misspecified, performance degrades gracefully to what would have happened if we optimized $\theta_{\text{spec}}$ by itself, so RLSP does not hurt. In addition, if $T$ is larger than it should be, then RLSP still tends to accurately infer parts of the reward that specify what not to do.

## 6   LIMITATIONS AND FUTURE WORK

**Summary.** Our key insight is that when a robot is deployed, the state that it observes has already been optimized to satisfy human preferences. This explains our preference for a policy that generally avoids side effects. We formalized this by assuming that Alice has been acting in the environment prior to the robot's deployment. We developed an algorithm, RLSP, that computes a MAP estimate of Alice's reward function. The robot then acts according to a tradeoff between Alice's reward function and the specified reward function. Our evaluation showed that information from the initial state can be used to successfully infer side effects to avoid as well as tasks to complete, though there are cases in which we cannot infer the relevant preferences. While we believe this is an important step forward, there is still much work to be done to make this accurate and practical.

**Realistic environments.** The primary avenue for future work is to scale to realistic environments, where we cannot enumerate states, we don't know dynamics, and the reward function may be nonlinear. This could be done by adapting existing IRL algorithms (Fu et al., 2017; Ho and Ermon, 2016; Finn et al., 2016). Unknown dynamics is particularly challenging, since we cannot learn dynamics from a single state observation. While acting in the environment, we would have to learn a dynamics model or an inverse dynamics model that can be used to simulate the past, and update the learned preferences as our model improves over time. Alternatively, if we use unsupervised skill learning (Achiam et al., 2018; Eysenbach et al., 2018; Nair et al., 2018) or exploration (Burda et al., 2018), or learn a goal-conditioned policy (Schaul et al., 2015; Andrychowicz et al., 2017), we could compare the explored states with the observed $s_0$.

**Hyperparameter choice.** While our evaluation showed that RLSP is reasonably robust to the choice of planning horizon $T$ and prior over $s_{-T}$, this may be specific to our gridworlds. In the real world, we often make long term hierarchical plans, and if we don't observe the entire plan (corresponding to a choice of T that is too small) it seems possible that we infer bad rewards, especially if we have an uninformative prior over $s_{-T}$. We do not know whether this will be a problem, and if so how bad it will be, and hope to investigate it in future work with more realistic environments.

**Conflicts between $\theta_{\text{spec}}$ and $\theta_{\text{Alice}}$.** RLSP allows us to infer $\theta_{\text{Alice}}$ from $s_0$, which we must somehow combine with $\theta_{\text{spec}}$ to produce a reward $\theta_{\text{final}}$ for the robot to optimize. $\theta_{\text{Alice}}$ will usually prefer the status quo of keeping the state similar to $s_0$, while $\theta_{\text{spec}}$ will probably incentivize some *change* to the state, leading to conflict. We traded off between the two by optimizing their sum, but future work could improve upon this. For example, $\theta_{\text{Alice}}$ could be decomposed into $\theta_{\text{Alice,task}}$, which says which task Alice is performing ("go to the black door"), and $\theta_{\text{frame}}$, which consists of the frame conditions ("don't break vases"). The robot then optimizes $\theta_{\text{frame}} + \lambda\theta_{\text{spec}}$. This requires some way of performing the decomposition. We could model the human as pursuing multiple different subgoals, or the environment as being created by multiple humans with different goals. $\theta_{\text{frame}}$ would be shared, while $\theta_{\text{task}}$ would vary, allowing us to distinguish between them. However, combination may not be the answer – instead, perhaps the robot ought to use the inferred reward to inform Alice of any conflicts and actively query her for more information, along the lines of Amin et al. (2017).

**Learning tasks to perform.** The apples and batteries environments demonstrate that RLSP can learn preferences that require the robot to actively perform a task. It is not clear that this is desirable, since the robot may perform an inferred task instead of the task Alice explicitly sets for it.

**Preferences that are not a result of human optimization.** While the initial state is optimized for human preferences, this may not be a result of *human* optimization, as assumed in this paper. For example, we prefer that the atmosphere contain oxygen for us to breathe. The atmosphere meets this preference *in spite of* human action, and so RLSP would not infer this preference. While this is of limited relevance for household robots, it may become important for more capable AI systems.

ACKNOWLEDGMENTS

We thank the researchers at the Center for Human Compatible AI for valuable feedback. This work was supported by the Open Philanthropy Project, AFOSR, and National Science Foundation Graduate Research Fellowship Grant No. DGE 1752814.

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

# A

Here, we derive an exact gradient for the maximum causal entropy distribution introduced in Ziebart et al. (2010), as the existing approximation is insufficient for our purposes. Given a trajectory $\tau_T = s_0 a_0 \ldots s_T a_T$, we seek the gradient $\nabla_\theta \ln p(\tau_T)$. We assume that the expert has been acting according to the maximum causal entropy IRL model given in Section 3 (where we have dropped $\theta$ from the notation for clarity):

$$\pi_t(a \mid s) = \exp(Q_t(s,a) - V_t(s)),$$

$$V_t(s) = \ln \sum_a \exp(Q_t(s,a)) \qquad \text{for } 1 \le t \le T,$$

$$Q_t(s,a) = \theta^T f(s) + \sum_{s'} \mathcal{T}(s' \mid s,a) V_{t+1}(s') \qquad \text{for } 1 \le t \le T,$$

$$V_{T+1}(s) = 0.$$

In the following, unless otherwise specified, all expectations over states and actions use the probability distribution over trajectories from the above model, starting from the state and action just prior. For example, $\mathbb{E}_{s'_T, a'_T}[X(s'_T, a'_T)] = \sum_{s'_T, a'_T} \mathcal{T}(s'_T \mid s_{T-1}, a_{T-1}) \pi_T(a'_T \mid s'_T) X(s'_T, a'_T)$. In addition, for all probability distributions over states and actions, we drop the dependence on $\theta$ for readability, so the probability of reaching state $s_T$ is written as $p(s_T)$ instead of $p(s_T \mid \theta)$.

First, we compute the gradient of $V_t(s)$. We have $\nabla_\theta V_{T+1}(s) = 0$, and for $0 \le t \le T$:

$$
\begin{aligned}
&\nabla_\theta V_t(s_t) \\
&= \nabla_\theta \ln \sum_{a'_t} \exp(Q_t(s_t, a'_t)) \\
&= \frac{1}{\exp(V_t(s_t))} \sum_{a'_t} \exp(Q_t(s_t, a'_t)) \nabla_\theta Q_t(s_t, a'_t) \\
&= \frac{1}{\exp(V_t(s_t))} \sum_{a'_t} \exp(Q_t(s_t, a'_t)) \nabla_\theta \left[ \theta^T f(s_t) + \mathbb{E}_{s'_{t+1} \sim \mathcal{T}(\cdot \mid s_t, a'_t)} \left[ V_{t+1}(s'_{t+1}) \right] \right] \\
&= \sum_{a'_t} \exp(Q_t(s_t, a'_t) - V_t(s_t)) \left[ f(s_t) + \mathbb{E}_{s'_{t+1} \sim \mathcal{T}(\cdot \mid s_t, a'_t)} \left[ \nabla_\theta V_{t+1}(s'_{t+1}) \right] \right] \\
&= \sum_{a'_t} \pi_t(a'_t \mid s_t) \left[ f(s_t) + \mathbb{E}_{s'_{t+1} \sim \mathcal{T}(\cdot \mid s_t, a'_t)} \left[ \nabla_\theta V_{t+1}(s'_{t+1}) \right] \right] \\
&= f(s_t) + \mathbb{E}_{a'_t, s'_{t+1}} \left[ \nabla_\theta V_{t+1}(s'_{t+1}) \right].
\end{aligned}
$$

Unrolling the recursion, we get that the gradient is the expected feature counts under the policy implied by $\theta$ from $s_t$ onwards, which we could prove using induction. Define:

$$\mathcal{F}_t(s_t) \equiv f(s_t) + \mathbb{E}_{a'_{t:T-1}, s'_{t+1:T}} \left[ \sum_{t'=t+1}^{T} f(s'_{t'}) \right].$$

Then we have:

$$\nabla_\theta V_t(s_t) = \mathcal{F}_t(s_t).$$

We can now calculate the gradient we actually care about:

$$\nabla_\theta \ln p(\tau_T)$$

$$= \nabla_\theta \left[ \ln p(s_0) + \sum_{t=0}^{T} \ln \pi_t(a_t \mid s_t) + \sum_{t=0}^{T-1} \ln \mathcal{T}(s_{t+1} \mid s_t, a_t) \right]$$

$$= \sum_{t=0}^{T} \nabla_\theta \ln \pi_t(a_t \mid s_t) \qquad\qquad\qquad\qquad \text{only } \pi_t \text{ depends on } \theta$$

$$= \sum_{t=0}^{T} \nabla_\theta \left[ Q_t(s_t, a_t) - V_t(s_t) \right]$$

$$= \sum_{t=0}^{T} \nabla_\theta \left[ \theta^T f(s_t) + \mathbb{E}_{s'_{t+1}} \left[ V_{t+1}(s'_{t+1}) \right] - V_t(s_t) \right]$$

$$= \sum_{t=0}^{T} \left( f(s_t) + \mathbb{E}_{s'_{t+1}} \left[ \nabla_\theta V_{t+1}(s'_{t+1}) \right] - \nabla_\theta V_t(s_t) \right).$$

The last term of the summation is $f(s_T) + \mathbb{E}_{s'_{T+1}} \left[ \nabla_\theta V_{T+1}(s'_{T+1}) \right] - \nabla_\theta V_T(s_T)$, which simplifies to $f(s_T) + 0 - \mathcal{F}_T(s_T) = f(s_T) - f(s_T) = 0$, so we can drop it. Thus, our gradient is:

$$\nabla_\theta \ln p(\tau_T) = \sum_{t=0}^{T-1} \left( f(s_t) + \mathbb{E}_{s'_{t+1}} \left[ \mathcal{F}_{t+1}(s'_{t+1}) \right] - \mathcal{F}_t(s_t) \right). \tag{6}$$

This is the gradient we will use in Appendix B, but a little more manipulation allows us to compare with the gradient in Ziebart et al. (2010). We reintroduce the terms that we cancelled above:

$$= \left( \sum_{t=0}^{T} f(s_t) \right) + \left( \sum_{t=0}^{T-1} \mathbb{E}_{s'_{t+1}} \left[ \mathcal{F}_{t+1}(s'_{t+1}) \right] \right) - \left( \mathcal{F}_0(s_0) + \sum_{t=0}^{T-1} \mathcal{F}_{t+1}(s_{t+1}) \right)$$

$$= \left( \sum_{t=0}^{T} f(s_t) \right) - \mathcal{F}_0(s_0) + \sum_{t=0}^{T-1} \left( \mathbb{E}_{s'_{t+1}} \left[ \mathcal{F}_{t+1}(s'_{t+1}) \right] - \mathcal{F}_{t+1}(s_{t+1}) \right).$$

Ziebart et al. (2010) states that the gradient is given by the expert policy feature expectations minus the learned policy feature expectations, and in practice uses the feature expectations from demonstrations to approximate the expert policy feature expectations. Assuming we have $N$ trajectories $\{\tau_i\}$, the gradient would be $\left( \frac{1}{N} \sum_i \sum_{t=0}^{T} f(s_{t,i}) \right) - \mathbb{E}_{s_0} \left[ \mathcal{F}_0(s_0) \right]$. The first term matches our first term exactly. Our second term matches the second term in the limit of sufficiently many trajectories, so that the starting states $s_0$ follow the distribution $p(s_0)$. Our third term converges to zero with sufficiently many trajectories, since any $s_t, a_t$ pair in a demonstration will be present sufficiently often that the empirical counts of $s_{t+1}$ will match the expected proportions prescribed by $\mathcal{T}(\cdot \mid s_t, a_t)$.

In a deterministic environment, we have $\mathcal{T}(s'_{t+1} \mid s_t, a_t) = 1[s'_{t+1} = s_{t+1}]$ since only one transition is possible. Thus, the third term is zero and even for one trajectory the gradient reduces to $\left( \sum_{t=0}^{T} f(s_t) \right) - \mathcal{F}_0(s_0)$. This differs from the gradient in Ziebart et al. (2010) only in that it computes feature expectations from the observed starting state $s_0$ instead of the MDP distribution over initial states $p(s_0)$.

In a stochastic environment, the third term need not be zero, and corrects for the "bias" in the observed states $s_{t+1}$. Intuitively, when the expert chose action $a_t$, she did not know which next state $s'_{t+1}$ would arise, but the first term of our gradient upweights the particular next state $s_{t+1}$ that we observed. The third term downweights the future value of the observed state and upweights the future value of all other states, all in proportion to their prior probability $\mathcal{T}(s'_{t+1} \mid s_t, a_t)$.

# B

This section provides a derivation of the gradient $\nabla_\theta \ln p(s_0)$, which is needed to solve $\text{argmax}_\theta \ln p(s_0)$ with gradient ascent. We provide the results first as a quick reference:

$$\nabla_\theta \ln p(s_0) = \frac{G_0(s_0)}{p(s_0)},$$

$$p(s_{t+1}) = \sum_{s_t,a_t} p(s_t)\pi_t(a_t \mid s_t)\mathcal{T}(s_{t+1} \mid s_t, a_t),$$

$$G_{t+1}(s_{t+1}) = \sum_{s_t,a_t} \mathcal{T}(s_{t+1} \mid s_t, a_t)\pi_t(a_t \mid s_t)\Big(p(s_t)g(s_t, a_t) + G_t(s_t)\Big),$$

$$g(s_t, a_t) \equiv f(s_t) + \mathbb{E}_{s'_{t+1}}\big[\mathcal{F}_{t+1}(s'_{t+1})\big] - \mathcal{F}_t(s_t),$$

$$\mathcal{F}_{t-1}(s_{t-1}) = f(s_{t-1}) + \sum_{a'_{t-1},s'_t} \pi_{t-1}(a'_{t-1} \mid s_{t-1})\mathcal{T}(s'_t \mid s_{t-1}, a'_{t-1})\mathcal{F}_t(s_t).$$

Base cases: first, $p(s_{-T})$ is given, second, $G_{-T}(s_{-T}) = 0$, and third, $\mathcal{F}_0(s_0) = f(s_0)$.

For the derivation, we start by expressing the gradient in terms of gradients of trajectories, so that we can use the result from Appendix A. Note that, by inspecting the final form of the gradient in Appendix A, we can see that $\nabla_\theta p(\tau_{-T:0})$ is independent of $a_0$. Then, we have:

$$\nabla_\theta \ln p(s_0) = \frac{1}{p(s_0)}\nabla_\theta p(s_0)$$

$$= \frac{1}{p(s_0)} \sum_{s_{-T:-1},a_{-T:0}} \nabla_\theta p(\tau_{-T:0})$$

$$= \frac{1}{p(s_0)} \sum_{s_{-T:-1},a_{-T:0}} p(\tau_{-T:0})\nabla_\theta \ln p(\tau_{-T:0})$$

$$= \frac{1}{p(s_0)} \sum_{s_{-T:-1},a_{-T:-1}} \left(p(\tau_{-T:-1}, s_0)\nabla_\theta \ln p(\tau_{-T:0})\left(\sum_{a_0}\pi_0(a_0 \mid s_0)\right)\right)$$

$$= \sum_{s_{-T:-1},a_{-T:-1}} p(\tau_{-T:-1} \mid s_0)\nabla_\theta \ln p(\tau_{-T:0}).$$

This has a nice interpretation – compute the gradient for each trajectory and take the weighted sum, where each weight is the probability of the trajectory given the evidence $s_0$ and current reward $\theta$.

We can rewrite the gradient in Equation 6 as $\nabla_\theta \ln p(\tau_T) = \sum_{t=0}^{T-1} g(s_t, a_t)$, where

$$g(s_t, a_t) \equiv f(s_t) + \mathbb{E}_{s'_{t+1}}\big[\mathcal{F}_{t+1}(s'_{t+1})\big] - \mathcal{F}_t(s_t).$$

We can now substitute this to get:

$$\nabla_\theta \ln p(s_0) = \sum_{s_{-T:-1},a_{-T:-1}} p(\tau_{-T:-1} \mid s_0)\left(\sum_{t=-T}^{-1} g(s_t, a_t)\right)$$

$$= \frac{1}{p(s_0)} \sum_{s_{-T:-1},a_{-T:-1}} \left[p(\tau_{-T:-1}, s_0)\sum_{t=-T}^{-1} g(s_t, a_t)\right]$$

$$= \frac{1}{p(s_0)} \sum_{s_{-T:-1},a_{-T:-1}} \left[p(\tau_{-T:-1}, s_0)\sum_{t=-T}^{-1} g(s_t, a_t)\right].$$

Note that we can compute $p(s_t)$ since we are given the distribution $p(s_{-T})$ and we can use the recursive rule $p(s_{t+1}) = \sum_{s_t,a_t} p(s_t)\pi_t(a_t \mid s_t)\mathcal{T}(s_{t+1} \mid s_t, a_t)$.

In order to compute $g(s_t, a_t)$ we need to compute $\mathcal{F}_t(s_t)$, which has base case $\mathcal{F}_0(s_0) = f(s_0)$ and recursive rule:

$$\mathcal{F}_{t-1}(s_{t-1})$$

$$= f(s_{t-1}) + \mathbb{E}_{a'_{t-1:-1},s'_{t:0}}\left[\sum_{t'=t}^{0} f(s'_{t'})\right]$$

$$= f(s_{t-1}) + \sum_{a'_{t-1},s'_t} \pi_{t-1}(a'_{t-1} \mid s_{t-1})\mathcal{T}(s'_t \mid s_{t-1}, a'_{t-1})\left[f(s'_t) + \mathbb{E}_{a'_{t:-1},s'_{t+1:0}}\left[\sum_{t'=t+1}^{0} f(s'_{t'})\right]\right]$$

$$= f(s_{t-1}) + \sum_{a'_{t-1},s'_t} \pi_{t-1}(a'_{t-1} \mid s_{t-1})\mathcal{T}(s'_t \mid s_{t-1}, a'_{t-1})\mathcal{F}_t(s_t).$$

For the remaining part of the gradient, define $G_t$ such that $\nabla_\theta \ln p(s_0) = \frac{G_0(s_0)}{p(s_0)}$:

$$G_t(s_t) \equiv \sum_{s_{-T:t-1},a_{-T:t-1}} \left[p(\tau_{-T:t-1}, s_t) \sum_{t'=-T}^{t-1} g(s_{t'}, a_{t'})\right].$$

We now derive a recursive relation for $G$:

$$G_{t+1}(s_{t+1})$$

$$= \sum_{s_{-T:t},a_{-T:t}} \left[p(\tau_{-T:t}, s_{t+1}) \sum_{t'=-T}^{t} g(s_{t'}, a_{t'})\right]$$

$$= \sum_{s_t,a_t} \sum_{s_{-T:t-1},a_{-T:t-1}} \mathcal{T}(s_{t+1} \mid s_t, a_t)\pi_t(a_t \mid s_t)p(\tau_{-T:t-1}, s_t)\left(g(s_t, a_t, s_{t+1}) + \sum_{t'=-T}^{t-1} g(s_{t'}, a_{t'})\right)$$

$$= \sum_{s_t,a_t} \left[\mathcal{T}(s_{t+1} \mid s_t, a_t)\pi_t(a_t \mid s_t)\left(\sum_{s_{-T:t-1},a_{-T:t-1}} p(\tau_{-T:t-1}, s_t)\right)g(s_t, a_t)\right]$$

$$+ \sum_{s_t,a_t} \left[\mathcal{T}(s_{t+1} \mid s_t, a_t)\pi_t(a_t \mid s_t) \sum_{s_{-T:t-1},a_{-T:t-1}} \left(p(\tau_{-T:t-1}, s_t) \sum_{t'=-T}^{t-1} g(s_{t'}, a_{t'})\right)\right]$$

$$= \sum_{s_t,a_t} \mathcal{T}(s_{t+1} \mid s_t, a_t)\pi_t(a_t \mid s_t)\left(p(s_t)g(s_t, a_t) + G_t(s_t)\right).$$

For the base case, note that

$$G_{-T+1}(s_{-T+1}) = \sum_{s_{-T},a_{-T}} [p(s_{-T}, a_{-T}, s_{-T+1})g(s_{-T}, a_{-T}, s_{-T+1})]$$

$$= \sum_{s_{-T},a_{-T}} \mathcal{T}(s_{-T+1} \mid s_{-T}, a_{-T})\pi_{-T}(a_{-T} \mid s_{-T})\left(p(s_{-T})g(s_{-T}, a_{-T}, s_{-T+1})\right).$$

Comparing this to the recursive rule, for the base case we can set $G_{-T}(s_{-T}) = 0$.

# C

Instead of estimating the MLE (or MAP if we have a prior) using RLSP, we could approximate the entire posterior distribution. One standard way to address the computational challenges involved with the continuous and high-dimensional nature of $\theta$ is to use MCMC sampling to sample from $p(\theta \mid s_0) \propto p(s_0 \mid \theta)p(\theta)$. The resulting algorithm resembles Bayesian IRL (Ramachandran and Amir, 2007) and is presented in Algorithm 1.

While this algorithm is less efficient and noisier than RLSP, it gives us an estimate of the full posterior distribution. In our experiments, we collapsed the full distribution into a point estimate by taking the mean. Initial experiments showed that the algorithm was slower and noisier than the gradient-based RLSP, so we did not test it further. However, in future work we could better leverage the full distribution, for example to create risk-averse policies, to identify features that are uncertain, or to identify features that are certain but conflict with the specified reward, after which we could actively query Alice for more information.

---

**Algorithm 1** MCMC sampling from the one state IRL posterior

---

**Require:** MDP $\mathcal{M}$, prior $p(\theta)$, step size $\delta$
1: $\theta \leftarrow$ random sample$(p(\theta))$
2: $\pi, V =$ soft value iteration$(\mathcal{M}, \theta)$
3: $p \leftarrow p(s_0 \mid \theta)p(\theta)$
4: **repeat**
5:     $\theta' \leftarrow$ random sample$(\mathcal{N}(\theta, \delta))$
6:     $\pi', V' =$ soft value iteration$(\mathcal{M}, \theta')$         $\triangleright$ The value function is initialized with $V$.
7:     $p' \leftarrow p(s_0 \mid \theta')p(\theta')$
8:     **if** random sample$(\text{Unif}(0, 1)) \leq \min(1, \frac{p'}{p})$ **then**
9:         $\theta \leftarrow \theta'; \;\; V \leftarrow V'$
10:    **end if**
11:    append $\theta$ to the list of samples
12: **until** have generated the desired number of samples

---

**Comparison of the methods for combining $\theta_{\text{spec}}$ and $\theta_{\text{H}}$**

Figure 4: Comparison of the Additive and Bayesian methods. We show how the percentage of true reward obtained by $\pi_{\text{RLSP}}$ varies as we change the tradeoff between $\theta_{\text{Alice}}$ and $\theta_{\text{spec}}$. The zero temperature case corresponds to traditional value iteration; this often leads to identical behavior and so the lines overlap. So, we also show the results when planning with soft value iteration, varying the softmax temperature, to introduce some noise into the policy. Overall, there is not much difference between the two methods. We did not include the Apples environment because $\theta_{\text{spec}}$ is uniformly zero and the Additive and Bayesian methods do exactly the same thing.

## D  COMBINING THE SPECIFIED REWARD WITH THE INFERRED REWARD

In Section 5, we evaluated RLSP by combining the reward it infers with a specified reward to get a final reward $\theta_{\text{final}} = \theta_{\text{Alice}} + \lambda\theta_{\text{spec}}$. As discussed in Section 6, the problem of combining $\theta_{\text{Alice}}$ and $\theta_{\text{spec}}$ is difficult, since the two rewards incentivize different behaviors and will conflict. The *Additive* method above is a simple way of trading off between the two.

Both RLSP and the sampling algorithm of Appendix C can incorporate a prior over $\theta$. Another way to combine the two rewards is to condition the prior on $\theta_{\text{spec}}$ before running the algorithms. In particular, we could replace our prior $P(\theta_{\text{Alice}})$ with a new prior $P(\theta_{\text{Alice}} \mid \theta_{\text{spec}})$, such as a Gaussian distribution centered at $\theta_{\text{spec}}$. When we use this prior, the reward returned by RLSP can be used as the final reward $\theta_{\text{final}}$.

It might seem like this is a principled *Bayesian* method that allows us to combine the two rewards. However, the conflict between the two reward functions still exists. In this formulation, it arises in the new prior $P(\theta_{\text{Alice}} \mid \theta_{\text{spec}})$. Modeling this as a Gaussian centered at $\theta_{\text{spec}}$ suggests that before knowing $s_0$, it seems likely that $\theta_{\text{Alice}}$ is very similar to $\theta_{\text{spec}}$. However, this is not true – Alice is probably providing the reward $\theta_{\text{spec}}$ to the robot so that it causes some *change* to the state that she has optimized, and so it will be *predictably* different from $\theta_{\text{spec}}$. On the other hand, we do need to put high probability on $\theta_{\text{spec}}$, since otherwise $\theta_{\text{final}}$ will not incentivize any of the behaviors that $\theta_{\text{spec}}$ did.

Nonetheless, this is another simple heuristic for how we might combine the two rewards, that manages the tradeoff between $\theta_{\text{spec}}$ and $\theta_{\text{Alice}}$. We compared the Additive and Bayesian methods by evaluating their robustness. We vary the parameter that controls the tradeoff and report the true reward obtained by $\pi_{\text{RLSP}}$, as a fraction of the expected true reward under the optimal policy. For the Bayesian method, we vary the standard deviation $\sigma$ of the Gaussian prior over $\theta_{\text{Alice}}$ that is centered at $\theta_{\text{spec}}$. For the Additive method, the natural choice would be to vary $\lambda$; however, in order to make the results more comparable, we instead set $\lambda = 1$ and vary the standard deviation of the Gaussian prior used while inferring $\theta_{\text{Alice}}$, which is centered at zero instead of at $\theta_{\text{spec}}$. A larger standard deviation allows $\theta_{\text{Alice}}$ to become larger in magnitude (since it is penalized less for deviating from the mean of zero reward), which effectively corresponds to a smaller $\lambda$.

While we typically create $\pi_{\text{RLSP}}$ using value iteration, this leads to deterministic policies with very sharp changes in behavior that make it hard to see differences between methods, and so we also show results with soft value iteration, which creates stochastic policies that vary more continuously. As demonstrated in Figure 4, our experiments show that overall the two methods perform very similarly, with some evidence that the Additive method is slightly more robust. The Additive method also has the benefit that it can be applied in situations where the inferred reward and specified reward are over different feature spaces, by creating the final reward $R_{\text{final}}(s) = \theta_{\text{Alice}}{}^T f_{\text{Alice}}(s) + \lambda R_{\text{spec}}(s)$.

