# OpenReview forum: "Preferences Implicit in the State of the World"
_ICLR.cc/2019/Conference_

### Official Review · AnonReviewer1 · 2018-10-20
**Strong paper; minor concerns**

**Rating:** 7
**Confidence:** 4

**Review:**

This paper considers the problem of inferring unspecified costs in an RL problem (e.g., inferring that vases in a room should not be broken). The primary insight is that the initial state of the environment conveys rich information about such unspecified costs since environments are often optimized for humans. The paper frames the problem of inferring unspecified costs from the initial condition as an inverse reinforcement learning (IRL) problem and applies the Maximum Causal Entropy IRL framework to solve this problem. Two methods are proposed for combining the inferred unspecified costs with specified costs. The efficacy of the proposed approach is demonstrated on a number of simulated examples.

Overall, I was impressed by this paper and I believe that it makes a strong contribution. The paper presents an interesting perspective on a relatively old problem (the frame problem in AI). The primary intuition of the paper (that the initial state conveys information about unspecified costs) and the framing of this problem in terms of IRL is novel. The simulated examples (while relatively simple in terms of the number of states and actions) are informative and demonstrate the strengths of the approach (and also some of the weaknesses; the paper is explicit about the current challenges). The paper is very clearly written and is easy to read.

My concerns are relatively minor:
- Perhaps the weakest bit of the paper is Section 5 (combining the specified reward with the inferred reward). As presented, the Additive method is somewhat hard to justify. However, the simulated results suggest that the Additive method performs slightly better than the Bayesian method. I would suggest either presenting a bit more intuition and justification for the Additive method or getting rid of this method altogether (since the results are not too different from the Bayesian method, which seems a bit more justifiable).
- One practical (and potentially important) question that the paper does not directly address is the problem of choosing the time horizon T (i.e., the time horizon for the past). In the standard IRL setting, it is reasonable to assume that the time horizon is given (since the demonstrations have an associated horizon). However, it is not entirely clear how to choose T in the setting considered in this paper. It is possible that if one chooses T to be too small, the inferred rewards will not be accurate (and one may have to look further back in the past to correctly infer rewards). A discussion of this issue and possible ways to choose T would be helpful.
- In Section 6.1 (baselines), the paper mentions that "while relative reachability makes use of known dynamics, it does not benefit from our handcoded featurization". Is it possible to modify the relative reachability method to also take advantage of the handcoded features, perhaps by considering dynamics over the feature space? If not, a sentence explaining that this is not straightforward would be helpful.
- In the related work section (and also in the introduction), I would recommend being more explicit about precisely what the differences are between the presented work and the approaches presented in (Krakovna et al. 2018) and (Turner, 2018). The paper is currently slightly vague about the differences.
- Currently, the title of the paper is a bit uninformative. On first reading the title, I expected a paper on control theory; the title makes no mention of unspecified costs, or reinforcement learning, or humans, etc. I believe that this is a good paper and that the paper would have more readers if the title was more inline with the content of the paper. Of course, this is at the discretion of the authors. My suggestion would be something along the lines of "Inferring Unspecified Rewards in RL from the Initial State".

Typos:
- Pg. 1, second paragraph, 3rd line: there is a placeholder for citations.
- Periods are missing at the end of equations.

---

> ### Author Response · Authors · 2018-11-22
> **Both methods of combining rewards are not very justifiable; and RLSP behaves reasonably with a misspecified horizon T**
>
> Thanks for the thorough review! We’re glad that you were impressed by the novelty and readability of the paper, and the usefulness of our evaluation. We respond to each of your concerns individually below.
>
> COMBINING REWARDS
>
> We actually find both the Bayesian and Additive methods to be unjustifiable. The general problem of combining θ_Alice and θ_spec is very difficult, and we expect that we will need a different formalism to solve it in a principled manner. The issue is that we have two sources of information about the best reward function for our robot -- θ_Alice inferred from the initial state s_0, and the reward specified by the designer θ_spec. θ_Alice will typically recommend keeping close to s_0, while the specified reward will recommend a change to s_0 (since we typically want our robots to change the environment somehow). So, these two sources are extremely likely to provide conflicting preference information. We are not sure how best to combine the two sources of information -- our best guess is that we should identify areas in which the two conflict, and ask Alice for clarification.
>
> In the case of the Bayesian method, this conflict arises in the prior P(θ_Alice | θ_spec). We have described this in more detail in the new Appendix D. We have also added a discussion of the general problem in Section 6, under the heading “Conflicts between θ_spec and θ_Alice”.
>
> Overall, we think this was a mistake in how we organized the paper. Our main contribution is our insight that the initial state contains preference information, as well as an example algorithm that can extract that preference information in the form of a reward function. We view both the Bayesian and Additive methods as unprincipled ways of combining rewards that were necessary for an evaluation, and have changed the paper to present them as such.
>
> (We also discuss related issues in our response to Reviewer 3.)
>
> CHOOSING THE TIME HORIZON
>
> The time horizon T is a fairly important parameter. However, even when T is very misspecified, RLSP ends up being uncertain about the reward function, and so we optimize something close to the specified reward θ_spec, so we are not any worse off. This also suggests that we could choose T by seeing which value of T leads to a distribution with minimal entropy. We have added an experiment with different values of T with more details; it is now Section 5.4.
>
> That said, this is only applicable to our simple gridworlds. In the real world, we often make long term hierarchical plans, and if we don’t observe the entire plan (corresponding to a choice of T that is too small) it seems possible that we infer bad rewards, especially if we have an uninformative prior over s_{-T}. We do not know whether this will be a problem, and if so how bad it will be, and hope to investigate it in future work with more realistic environments. We have added a discussion on this issue to the limitations section.
>
> RELATIVE REACHABILITY
>
> It is possible to define a version of relative reachability that operates in feature space instead of state space, effectively collapsing states with the same features into a single entity. This has the benefit of not capturing any of the irreversibilities that the featurization does not capture (which presumably humans don’t care about), but this doesn’t matter for our gridworlds, so it doesn’t make a difference to our experiments.
>
> CLARITY
>
> (Krakovna et al. 2018) and (Turner, 2018) are both impact measures -- they penalize any high impact action, including ones that we actually want. In contrast, we can distinguish between impact that humans do and don’t care about. We have clarified this in the introduction and related work.
>
> Thanks for pointing out the typos, we have fixed them now. We also appreciate your pointing out that the title was a bit uninformative, and have now added the word “preferences” to the title, such that the new one is “The Implicit Preference Information in an Initial State”. We want to avoid terms like “Inferring rewards” in order to keep the emphasis on the idea that the initial state has preference information (which we hope researchers will build on), instead of the particular algorithm that we propose (which we hope will be superseded by one that makes fewer assumptions).

---

### Official Review · AnonReviewer3 · 2018-11-04
**An interesting take on combining explicit and inferred reward functions, but limited by unresolved questions and few quantitative results**

**Rating:** 6
**Confidence:** 3

**Review:**

The authors propose to augment the explicitly stated reward function of an RL agent with auxiliary rewards/costs inferred from the initial state and a model of the state dynamics.  Intuitively, the fact that a vase precariously placed in the center of the room remains intact suggests that it is a precious object that should be handled with care, even though the reward function may not explicitly say so.  Technically, implicit rewards like these are inferred via inverse reinforcement learning: the agent (e.g. robot) first estimates the most likely reward functions to have guided existing agents (e.g. humans) by integrating over all possible state-action paths that could have led to the initial condition and evaluating their probability under different rewards (and hence different optimal policies).  The proposal is clever, but there are some philosophical hurdles to overcome and the experimental results offer little quantitative evidence to support this idea.

In my view, the biggest challenge is how to balance explicitly stated rewards with those inferred from the initial condition.  Section 5 briefly addresses this question, but essentially capitulates by saying, "This trade-off is inevitable given our problem formulation, since we have two sources of information...and they will conflict in some cases."  I fear this conflict may be the rule rather than the exception.  For example, when I deploy my brand new dish-washing robot on my sink full of dirty dishes, my instructions to clean up will be in direct conflict with my past self's actions (or lack thereof).  How is the agent to know how strongly to adhere to the stated goals and when to deviate?  One possible solution is to only allow the inferred reward to affect features that are not explicitly included in the specified reward.  Neither the Additive nor the Bayesian combination methods have this property though.

The technical presentation could use some improvement.  The preliminaries in Section 3 do a decent job of introducing MDPs and IRL, but stop short of saying how the objective function for MCEIRL is actually computed.  Specifically, theta does not appear on the right hand side of Eq (1); implicitly, pi is a function of theta that is estimated, presumably, via value or policy iteration.  The marginal probability of the initial state and its gradients presented in Section 4.1 are the main technical contribution of the paper, but most of the key details are deferred to the appendix or referenced to Ziebart (2010).  For example, the dynamic programming algorithm for computing Eq (3) and the expectations over state-action paths in Eq (5) could use more discussion in the main text, as could some elements of the derivation of Eq (5).

The experimental results are presented primarily in words (e.g. "\pi_spec walks over the vase while \pi_deviation and \pi_reachability both avoid it.").  It would be helpful to see the resulting paths taken by the various agents, or even better, to see their learned reward functions alongside the true reward functions.  The only quantitative results are those in Figure 3, and unfortunately they are a bit confusing.  Why would we expect non-monotonic rewards at some temperatures?  Moreover, why are some reward "percentages" negative?

The idea of leveraging the initial state for augmenting the reward function is clever, but there are a few shortcomings of the current paper.  There are basic concerns about how implicit and explicit rewards can be combined, and the technical presentation needs some improvement.  Most importantly, the experimental results do not show enough quantitative evidence of how the proposed method performs.

[UPDATE] I appreciate the authors' detailed response and revisions to the paper.  I've updated my score accordingly.

---

> ### Author Response · Authors · 2018-11-22
> **We agree that combining the two rewards is tricky, we don't claim to solve it and have reorganized the paper to make that clear**
>
> Thanks for the detailed and thoughtful review! We’re happy that you appreciated the proposal to infer implicit rewards by integrating over trajectories that could have led to the initial state under various reward functions. While we very much agree with the technical content of your review (in hindsight we would have had the same reaction to reading the paper ourselves, and it has helped us present the paper better), we disagree on its implication on the merit of the paper.
>
> COMBINING INFERRED AND SPECIFIED REWARD
>
> We agree that figuring out the right way to combine the inferred and specified rewards is tricky. It is indeed the rule for them to conflict -- the intended meaning of the sentence you quoted was that in almost all environments the rewards will conflict in some states, but this was ambiguous, and we have updated the paper to fix this. But what we ask you to consider is that getting access to the implicit reward to begin with is important, and that is our contribution. Indeed, much work will need to happen to sort out exactly how to use it, but we see our main contribution as the inference of that implicit reward itself, and we think that is a really important step that opens up a rich area of investigation.
>
> From that perspective, we do not claim that our heuristics for the combination are the final solution in any way. There are many methods that could make this work in our simplified gridworlds, which would give results nominally better than the Bayesian and Additive methods. For example, as you mentioned we could have the inferred reward only affect features that were unspecified in the explicitly specified reward. Another possibility is to try to decouple frame conditions (things that should remain the same) from the task that the human is trying to perform -- for example, we could consider all states from which the current state is reachable, treat all of those states as goals, and take the average inferred reward from each of them. This will end up “averaging out” the goal that the human was aiming for, while still inferring the negative rewards on irreversible actions that the human didn’t take (like breaking vases). We wrote a quick implementation of this and it does improve the inferred reward on environments meant to test frame conditions (room with vase and toy train).
>
> However, our key aim with this work is to show conceptually that the initial state contains preference information that can be learned, which we are quite confident will generalize to realistic environments as well (as we discuss briefly in Section 6). Approaches like “averaging” over all possible states from which s_0 is reachable are much more speculative -- while they may improve results on these gridworlds, we would not bet on it working in more complex environments. Similarly, we may not have the luxury of a hand-designed feature space in order to have the inferred reward only affect features that were unspecified.
>
> Combining inferred and explicit rewards might require some new formalism -- for example, we may model the human as pursuing multiple different subgoals, or perhaps we model them as acting randomly subject to some constraints (and we infer the constraints from the initial state), or we could model the environment as being created by multiple humans with similar but not identical goals. Ultimately, our honest take on this is that combination is not really the answer -- instead, the robot ought to use the inferred reward to actively query the human for more information. We plan to investigate this in future work, and have incorporated some of this discussion in the paper, in Section 6.
>
> The reason we had a way to combine rewards at all was because we were faced with the task of how to evaluate this proposal. We could have inspected the learned rewards qualitatively, but this seemed quite error-prone and subjective. A better evaluation would look at the behavior incentivized by the inferred reward function, but in most cases an inferred reward function would take no action and leave the state as it is. So, we needed to evaluate by considering an explicit, misspecified reward function, and seeing whether the inferred reward function could correct it to get the right behavior. We have moved the description of the combination methods to the experiments section to emphasize this.

---

> ### Author Response · Authors · 2018-11-22
> **Quantitative metrics are not appropriate for this domain, but we agree that we should show trajectories**
>
> EXPERIMENTAL EVALUATION
>
> Thanks for the suggestion to show the paths and rewards taken by the various agents -- we have updated the paper with a figure showing this.
>
> We chose not to present quantitative results for that section because the most reasonable quantitative metric in our setting seemed dishonest to us. We’re referring to the metric of the fraction of max reward obtained when replanning using the inferred reward. Intuitively, this seems like it captures our notion of “did we infer the right reward”. Unfortunately, while the optimal behavior for a reward function is invariant to reward shaping such as adding a constant, this quantitative metric is not. This means that we could change the specified reward function and the form of the prior to get the results we wanted. For example, by adding constants to the specified reward, we could get our baselines to show a fraction of either 10% or 90%, while our method remains at 100%. (RLSP almost always finds the optimal trajectory, except in the room with far away vase environment, and so would usually be at 100%.)
>
> Ultimately, what we actually care about is the behavior incentivized by the reward function. If we didn’t have any better metric, we would use the fraction of max reward metric. However, in the simple setting of gridworlds with deterministic planning, it is actually feasible to describe or show the behavior itself, so we decided to do that, since it is more informative without being overwhelming.
>
> For the quantitative results comparing the Additive and Bayesian methods of combining rewards, the reward percentages could be negative because it was possible for the policies to get negative reward (eg. by breaking vases or trains). We decided to add a constant to the reward function to force it to be non-negative, which doesn't change the optimal trajectory or inferred reward, but does allow our reward percentages to be non-negative and so less confusing.
>
> We’re not quite sure what you mean by the results being non-monotonic. Note that the x-axis in that figure is the standard deviation of the prior over θ -- essentially this is meant to vary the parameter that controls the tradeoff between θ_spec and θ_Alice. The optimal reward will come at some intermediate value of tradeoff between the two rewards, and so we shouldn’t expect a monotonic curve.
>
> TECHNICAL PRESENTATION
>
> We appreciate the points on technical presentation. The policy pi in Eq. (1) is computed using value iteration, which is dependent on the reward parameters theta. We have updated the equations in that section to show the dependence on theta.
>
> We are playing around with ways to integrate more detail in the main text, but are unsure what to take out. Currently, our thinking is that any future work in more realistic environments is more likely to draw on the ideas in this paper rather than the particular technical details, and so we have focused on the ideas in the main paper. We have already moved the details about the Bayesian method of combining rewards and the experiments on it to the appendix, since as we mentioned above, we view the combination of rewards more as a necessity for evaluation than one of our contributions. However, we are still over eight pages currently. Do you have suggestions on things to remove?

---

### Official Review · AnonReviewer2 · 2018-11-04
**Original formulation of initial state exploration for robot action optimisation - Reinforcement Learning**

**Rating:** 7
**Confidence:** 3

**Review:**

The framework of this work is Reinforcement Learning (RL) optimisation. The data consists of states of the space where the action takes place. Actions are possible, and they lead to possible transitions in the state space. A reward function assesses how adequate a state space is.
The main originality of the work is to use the initial state as a key information about the features that translate many desired state of background objects in a scene. An algorithm is built to make use of this information to build an ad hoc reward function, which specifies a good landscape of desired vs non-desired states of the space. An empirical evaluation of the introduced method is presented. It is rich and interesting, although hard to fully grasp for a non-expert.

Key questions/remarks:
 - how different is your approach to a Bayesian approach with the combination of a likelihood (~reward) and prior (~initial state analysis) into a posterior distribution of the space? This seems to be the case in Section 5, where your alternative formulation clearly resembles a Lasso approach (which can be cast in a Bayesian framework).
 - I quite like your decomposition of your ideas into many titled paragraphs. The drawback is that there is sometimes a lack of connections between the many ideas you combine. A would see a big figure in the form of a map as a central contribution of your work to explain the different bits. Still, I appreciate the effort to have a synthetic contribution!

Small remarks:
 - the abstract could be improved to provide an easier reading experience
 - first time IRL on p2 is mentioned, without a prior explanation of the acronym
 - the world is already optimised for human preferences: yes and no, this is one of your (strong?) assumptions. The robot could well move the vase to a location which is acceptable. Or put it back.
 - on p3, beg.  of Section 3, explain the decomposition of r(s) = \theta^{T}f(s).
 - in IRL paragraph: say the elements of \tau_{i} are s.t. the transitions need be possible.
 - p8 'access to a simulator': what can be simulated if very little is know about the background, but via an initial state?
 - past point of the discussion: I simply don't get it!?!

---

> ### Author Response · Authors · 2018-11-22
> **Miscellaneous answers**
>
> Thanks for the positive review! We’re glad that you found the work original and the empirical evaluation rich and interesting. We respond to individual points below:
>
> KEY QUESTIONS/REMARKS:
>
> For the section on combining rewards, certainly for our first approach (the one we call “Bayesian”) we agree with your characterization of it as a Bayesian approach. We prefer to think of the specified reward as the prior and the initial state analysis as the likelihood, but of course you could view it the other way as well.
>
> It does seem possible that the Additive approach that we ultimately use could also be reformulated as a Bayesian approach, where we use a Laplace prior (which is equivalent to Lasso L1 regularization). This would explain why the two methods perform so similarly. However, we have not investigated this in detail (because we view these techniques as unprincipled methods that were necessary for an evaluation, see our response to Reviewers 1 and 3 for more details), and so this should be taken as speculation on our part.
>
> We’re happy you like the organization of our paper, but we’re not sure exactly what you mean by “a big figure in the form of a map as a central contribution of your work”, could you expand more on this? We were hoping that Figure 1 would serve as the main explanation of our work (though of course it does not capture everything).
>
> SMALL REMARKS
>
> We have updated the paper to address the first, second, fourth and fifth points. (We’re not sure what in particular about the abstract would make an easier reading experience, so please do tell us if there’s something else we can improve.)
>
> For your third remark, we’re not sure we understand what you mean here. We do not think that our key assumption is strong: certainly we humans have been changing the world to meet our preferences for the past thousands of years. We agree that the robot could move the vase to an acceptable location or put it back, but don’t see the relevance to our assumption. Our best guess is that you think that our assumption means that the state of the world should never change at all, and so we would never allow the robot to move the vase. However, if the robot considers the two possible human reward functions, “don’t break the vase” and “don’t move the vase from this particular location”, both of these make the observed state quite likely, and so the robot will be uncertain between these two reward functions. In addition, in a realistic environment there are likely to be many vases that are all unbroken, and the reward function “don’t break the vase” will be a much simpler explanation of the initial state than the reward function “The first vase must be at this particular location, and the second vase must be at this location, and …”.
>
> For your sixth remark about “access to a simulator”, are you asking what can be done in the case where we don’t have a simulator? Nearly all applications of deep RL require access to a simulator -- even applications that work on real robots will typically train in simulation and then transfer to the real world (though some exceptions do exist). If we do not have access to a simulator, and only have access to the initial state, and do not know dynamics, we don’t know how to make our method work -- but nearly all existing methods do not work in such a setting.
>
> For the seventh point, we have updated the paper slightly to make it clearer, but we provide a longer explanation here. The key insight of our paper is that the world is already optimized for human preferences. Our algorithm makes another assumption -- that the reason that the world satisfies human preferences is because a human made it that way by acting over the past T timesteps. However, there are some preferences that are automatically satisfied independent of human action, and we won’t be able to infer these preferences. For example, in order for us to stay alive, we need the atmosphere to contain oxygen. However, we humans are not responsible for the atmosphere containing oxygen -- in fact, regardless of what we could have done in our history so far, we could not have prevented the atmosphere from containing oxygen. As a result, RLSP will note that no matter what reward function we have, the atmosphere will always contain oxygen, and so the fact that it observes an initial state with oxygen is unsurprising, and tells it nothing about whether humans like or dislike oxygen.

---

### Official Review · AnonReviewer4 · 2018-11-09
**An interesting idea but less convincing experimental results**

**Rating:** 6
**Confidence:** 4

**Review:**

This work proposes a way to infer the implicit information in the initial state using IRL and combine the inferred reward with a specified reward to achieve better performance in a few simulated environments where the specified reward is not sufficient to solve the task. The main novelty of this work is to reformulate the Maximum Causal Entropy IRL objective using just the initial state as the end state of an expert trajectory to infer the underlying preference. Overall the proposed approach is impressive and the intuition behind the paper is novel and easy to understand.

My main concerns are the following:
- All the simulated experiments are able to demonstrate the effectiveness of the method, though they seem to be a bit too simplistic, e.g. known dynamics. As mentioned in Section 7, more real-environment experiments would make this method a lot stronger.
- The way of choosing the distribution s_{-T} seems to require some sort of human preference, e.g. in the apple collection case,  s_{-T} has to be sampled from the distribution where there's no apple in the basket in order to make the algorithm to work. This assumption seems to make the implicit information of the initial state not so *implicit*. Besides, it's unclear how to choose the horizon T. It would be interesting to see how the value of T affects the performance.

---

> ### Author Response · Authors · 2018-11-22
> **Experiments are meant to evaluate the conceptual idea, not prove that it can be applied immediately; RLSP is somewhat robust to hyperparameter choice**
>
> Thanks for the cogent review! We’re pleased that you found our approach impressive and the intuition novel and easy to understand. We respond to your two main concerns below.
>
> SIMPLISTIC EXPERIMENTS
>
> We agree that a convincing demonstration of the effectiveness of the method would require experiments on more realistic environments. However, with realistic environments come many problems that are unrelated to the conceptual core of the idea. If we run an experiment and it fails to learn preferences, is this because of a problem with the idea, or with the implementation? Deep RL is notoriously hard to implement correctly, and even when it is working, it is hard to tell how much different components are helping. We are proposing a new problem to solve, where we aim to learn preferences from a single state. We believe that it is worthwhile to see how far we can get in a simplified domain to establish what is conceptually possible, before delving into the problem of getting the method to work in more realistic settings.
>
> DISTRIBUTION OVER s_{-T}
>
> After we submitted this paper, we continued to investigate the results with a uniform prior over s_{-T}, because they were very counterintuitive to us. We found that the gradient from Ziebart (2010) that we were using was an approximation that works well with large amounts of data, but leads to significant errors when our “dataset” consists of a single state. We derived an exact formula for the gradient (now in Appendix A), and with this new gradient, the results with a uniform prior are more in line with what we expected: the inferred rewards are qualitatively the same as with known s_{-T}, and they still lead to good behavior. Now, in the apple collection case, a uniform prior over s_{-T} (without the knowledge that there are no apples in the basket) does lead to good apple harvesting behavior.
>
> Of course, more generally there certainly is a dependence on the distribution over s_{-T}. In fact, we would go further and say that there is a dependence on the dynamics of the environment, the featurization that we get to use, and the horizon T. Sometimes the initial state won’t have all of the preference information, and we won’t infer perfect preferences. For example, suppose it were possible to put apples from the basket back on the trees. Then RLSP would consider both the case where we start with four apples and put apples back on the tree, and the case where we start with zero apples and put apples in the basket, and these would balance out, leaving it uncertain about the reward function. We expect that when the initial state doesn’t have enough information to infer preferences, RLSP will be more uncertain about the reward. Note that even when RLSP is maximally uncertain about the reward, the overall behavior will be to optimize the specified reward, which is what we were going to do anyway, so our method degrades gracefully to the performance we would have gotten anyway.
>
> We do believe that the results in this paper suggest that our method can be fairly robust to the choice of prior over s_{-T}, and it can be much easier to design a prior over s_{-T} than it is to write down a correct reward function or to provide demonstrations. Any environment with a simulator (i.e. nearly everything considered in deep RL currently) typically also comes with an initial state for that simulator where humans have not yet acted -- we could consider that as our s_{-T}. For example, in Minecraft, humans have built many structures within the Minecraft world, but we can still easily initialize a fresh Minecraft world and simulate from there.
>
> CHOOSING THE HORIZON T
>
> We have added an experiment on this (Section 5.4) that suggests that choosing T is important for inferring things to do (as in the apples environment) but not as important for inferring what not to do (such as not breaking vases).

---

### Comment · AnonReviewer1 · 2018-11-23
**Still advocating for acceptance of the paper**

I have carefully gone through all the other reviews and the authors' response to them. I have also gone through some of the revisions made to the paper. The authors have added numerical experiments to address one of my main technical concerns (choosing the time horizon T) and have also added a large amount of useful discussion regarding combination of the specified rewards and inferred rewards (which other reviewers pointed out as well).

Overall, the paper introduces a novel and interesting idea (inferring preferences from the initial state of an environment), which I see as the primary contribution of the paper. The paper proposes algorithms that implement this idea and a number of experiments that support the idea. The authors are very clear about the limitations of the work and have a significant amount of discussion on how these may be addressed.  I believe that the ideas in the paper will lead to significant follow-up work (both from the authors themselves and others). Overall, I still believe that this paper makes a strong contribution and am happy to advocate for its acceptance.

---

### Meta-Review · Area_Chair1 · 2018-12-14
**Interesting idea and setup, although technical contribution is somewhat limited**

**Confidence:** 3
**Recommendation:** Accept (Poster)

**Metareview:**

The paper proposes to take advantage of implicit preferential information in a single state, to design auxiliary reward functions that can be combined with the standard RL reward function.  The motivation is to use the implicit information to infer signals that might not have been included in the reward function.  The paper has some nice ideas and is quite novel.  A new algorithm is developed, and is supported by proof-of-concept experiments.

Overall, the paper is a nice and novel contribution.  But reviewers point out several limitations.  The biggest one seems to be related to the problem setup: how to combine inferred reward and the given reward, especially when they are in conflict with each other.  A discussion of multi-objective RL might be in place.